# Finite Sample Analysis of the GTD Policy Evaluation Algorithms in Markov Setting

**Yue Wang** *
School of Science
Beijing Jiaotong University
11271012@bjtu.edu.cn

**Wei Chen**
Microsoft Research
wche@microsoft.com

**Yuting Liu**
School of Science
Beijing Jiaotong University
ytliu@bjtu.edu.cn

**Zhi-Ming Ma**
Academy of Mathematics and Systems Science
Chinese Academy of Sciences
mazm@amt.ac.cn

**Tie-Yan Liu**
Microsoft Research
Tie-Yan.Liu@microsoft.com

## Abstract

In reinforcement learning (RL) , one of the key components is policy evaluation, which aims to estimate the value function (i.e., expected long-term accumulated reward) of a policy. With a good policy evaluation method, the RL algorithms will estimate the value function more accurately and find a better policy. When the state space is large or continuous *Gradient-based Temporal Difference(GTD)* policy evaluation algorithms with linear function approximation are widely used. Considering that the collection of the evaluation data is both time and reward consuming, a clear understanding of the finite sample performance of the policy evaluation algorithms is very important to reinforcement learning. Under the assumption that data are i.i.d. generated, previous work provided the finite sample analysis of the GTD algorithms with constant step size by converting them into convex-concave saddle point problems. However, it is well-known that, the data are generated from Markov processes rather than i.i.d. in RL problems.. In this paper, in the realistic Markov setting, we derive the finite sample bounds for the general convex-concave saddle point problems, and hence for the GTD algorithms. We have the following discussions based on our bounds. (1) With variants of step size, GTD algorithms converge. (2) The convergence rate is determined by the step size, with the mixing time of the Markov process as the coefficient. The faster the Markov processes mix, the faster the convergence. (3) We explain that the experience replay trick is effective by improving the mixing property of the Markov process. To the best of our knowledge, our analysis is the first to provide finite sample bounds for the GTD algorithms in Markov setting.

## 1   Introduction

Reinforcement Learning (RL) (Sutton and Barto [1998]) technologies are very powerful to learn how to interact with environments, and has variants of important applications, such as robotics, computer games and so on (Kober et al. [2013], Mnih et al. [2015], Silver et al. [2016], Bahdanau et al. [2016]).

In RL problem, an agent observes the current state, takes an action following a policy at the current state, receives a reward from the environment, and the environment transits to the next state in Markov, and again repeats these steps. The goal of the RL algorithms is to find the optimal policy which

leads to the maximum long-term reward. The value function of a fixed policy for a state is defined as the expected long-term accumulated reward the agent would receive by following the fixed policy starting from this state. Policy evaluation aims to accurately estimate the value of all states under a given policy, which is a key component in RL (Sutton and Barto [1998], Dann et al. [2014]). A better policy evaluation method will help us to better improve the current policy and find the optimal policy.

When the state space is large or continuous, it is inefficient to represent the value function over all the states by a look-up table. A common approach is to extract features for states and use parameterized function over the feature space to approximate the value function. In applications, there are linear approximation and non-linear approximation (e.g. neural networks) to the value function. In this paper, we will focus on the linear approximation (Sutton et al. [2009a],Sutton et al. [2009b],Liu et al. [2015]). Leveraging the localization technique in Bhatnagar et al. [2009], the results can be generated into non-linear cases with extra efforts. We leave it as future work.

In policy evaluation with linear approximation, there were substantial work for the temporal-difference (TD) method, which uses the Bellman equation to update the value function during the learning process (Sutton [1988],Tsitsiklis et al. [1997]). Recently, Sutton et al. [2009a] Sutton et al. [2009b] have proposed *Gradient-based Temporal Difference (GTD)* algorithms which use gradient information of the error from the Bellman equation to update the value function. It is shown that, GTD algorithms have achieved the lower-bound of the storage and computation complexity, making them powerful to handle high dimensional big data. Therefore, now GTD algorithms are widely used in policy evaluation problems and the policy evaluation step in practical RL algorithms (Bhatnagar et al. [2009],Silver et al. [2014]).

However, we don't have sufficient theory to tell us about the finite sample performance of the GTD algorithms. To be specific, will the evaluation process converge with the increasing of the number of the samples? If yes, how many samples we need to get a target evaluation error? Will the step size in GTD algorithms influence the finite sample error? How to explain the effectiveness of the practical tricks, such as experience replay? Considering that the collection of the evaluation data is very likely to be both time and reward consuming, to get a clear understanding of the finite sample performance of the GTD algorithms is very important to the efficiency of policy evaluation and the entire RL algorithms.

Previous work (Liu et al. [2015]) converted the objective function of GTD algorithms into a convex-concave saddle problem and conducted the finite sample analysis for GTD with constant step size under the assumption that data are i.i.d. generated. However, in RL problem, the date are generated from an agent who interacts with the environment step by step, and the state will transit in Markov as introduced previously. As a result, the data are generated from a Markov process but not i.i.d.. In addition, the work did not study the decreasing step size, which are also commonly-used in many gradient based algorithms(Sutton et al. [2009a],Sutton et al. [2009b],Yu [2015]). Thus, the results from previous work cannot provide satisfactory answers to the above questions for the finite sample performance of the GTD algorithms.

In this paper, we perform the finite sample analysis for the GTD algorithms in the more realistic Markov setting. To achieve the goal, first of all, same with Liu et al. [2015], we consider the stochastic gradient descent algorithms of the general convex-concave saddle point problems, which include the GTD algorithms. The optimality of the solution is measured by the primal-dual gap (Liu et al. [2015], Nemirovski et al. [2009]). The finite sample analysis for convex-concave optimization in Markov setting is challenging. On one hand, in Markov setting, the non-i.i.d. sampled gradients are no longer unbiased estimation of the gradients. Thus, the proof technique for the convergence of convex-concave problem in i.i.d. setting cannot be applied. On the other hand, although SGD converge for convex optimization problem with the Markov gradients, it is much more difficult to obtain the same results in the more complex convex-concave optimization problem.

To overcome the challenge, we design a novel decomposition of the error function (i.e. Eqn (3.1)). The intuition of the decomposition and key techniques are as follows: (1) Although samples are not i.i.d., for large enough $\tau$, the sample at time $t + \tau$ is "nearly independent" of the sample at time $t$, and its distribution is "very close" to the stationary distribution. (2) We split the random variables in the objective related to $\mathbb{E}$ operator and the variables related to $\max$ operator into different terms in order to control them respectively. It is non-trivial, and we construct a sequence of auxiliary random variables to do so. (3) All constructions above need to be carefully considered the measurable issues

in the Markov setting. (4) We construct new martingale difference sequences and apply Azuma's inequality to derive the high-probability bound from the in-expectation bound.

By using the above techniques, we prove a novel finite sample bound for the convex-concave saddle point problem. Considering the GTD algorithms are specific convex-concave saddle point optimization methods, we finally obtained the finite sample bounds for the GTD algorithms, in the realistic Markov setting for RL. To the best of our knowledge, our analysis is the first to provide finite sample bounds for the GTD algorithms in Markov setting.

We have the following discussions based on our finite sample bounds.

1. GTD algorithms do converge, under a flexible condition on the step size, i.e. $\sum_{t=1}^{T} \alpha_t \to \infty$, $\frac{\sum_{t=1}^{T} \alpha_t^2}{\sum_{t=1}^{T} \alpha_t} < \infty$, as $T \to \infty$, where $\alpha_t$ is the step size. Most of step sizes used in practice satisfy this condition.

2. The convergence rate is $\mathcal{O}\left(\sqrt{(1+\tau(\eta))\frac{\sum_{t=1}^{T} \alpha_t^2}{\sum_{t=1}^{T} \alpha_t}} + \frac{\sqrt{\tau(\eta)\log(\frac{\tau(\eta)}{\delta})\sum_{t=1}^{T} \alpha_t^2}}{\sum_{t=1}^{T} \alpha_t}\right)$, where $\tau(\eta)$ is the mixing time of the Markov process, and $\eta$ is a constant. Different step sizes will lead to different convergence rates.

3. The experience replay trick is effective, since it can improve the mixing property of the Markov process.

Finally, we conduct simulation experiments to verify our theoretical finding. All the conclusions from the analysis are consistent with our empirical observations.

## 2 Preliminaries

In this section, we briefly introduce the GTD algorithms and related works.

### 2.1 Gradient-based TD algorithms

Consider the reinforcement learning problem with Markov decision process(MDP) $(\mathcal{S}, \mathcal{A}, P, R, \gamma)$, where $\mathcal{S}$ is the state space, $\mathcal{A}$ is the action space, $P = \{P_{s,s'}^a; s, s' \in \mathcal{S}, a \in \mathcal{A}\}$ is the transition matrix and $P_{s,s'}^a$ is the transition probability from state $s$ to state $s'$ after taking action $a$, $R = \{R(s,a); s \in \mathcal{S}, a \in \mathcal{A}$ is the reward function and $R(s,a)$ is the reward received at state $s$ if taking action $a$, and $0 < \gamma < 1$ is the discount factor. A policy function $\mu : \mathcal{A} \times \mathcal{S} \to [0, 1]$ indicates the probability to take each action at each state. Value function for policy $\mu$ is defined as: $V^{\mu}(s) \triangleq E\left[\sum_{t=0}^{\infty} \gamma^t R(s_t, a_t)|s_0 = s, \mu\right]$.

In order to perform policy evaluation in a large state space, states are represented by a feature vector $\phi(s) \in \mathbb{R}^d$, and a linear function $\hat{v}(s) = \phi(s)^{\top}\theta$ is used to approximate the value function. The evaluation error is defined as $\|V(s) - \hat{v}(s)\|_{s \sim \pi}$, which can be decomposed into approximation error and estimation error. In this paper, we will focus on the estimation error with linear function approximation.

As we know, the value function in RL satisfies the following Bellman equation: $V^{\mu}(s) = \mathbb{E}_{\mu,P}\left[R(s_t, a_t) + \gamma V^{\mu}(s_{t+1})|s_t = s\right] \triangleq T^{\mu}V^{\mu}(s)$, where $T^{\mu}$ is called Bellman operator for policy $\mu$. Gradient-based TD (GTD) algorithms (including GTD and GTD2) proposed by Sutton et al. [2009a] and Sutton et al. [2009b] update the approximated value function by minimizing the objective function related to Bellman equation errors, i.e., the norm of the expected TD update (NEU) and mean-square projected Bellman error (MSPBE) respectively(Maei [2011],Liu et al. [2015]) ,

$$GTD: \quad J_{NEU}(\theta) = \|\Phi^{\top}K(T^{\mu}\hat{v} - \hat{v})\|^2 \tag{2.1}$$

$$GTD2: \quad J_{MSPBE}(\theta) = \|\hat{v} - \mathcal{P}T^{\mu}\hat{v}\| = \|\Phi^{\top}K(T^{\mu}\hat{v} - \hat{v})\|_{C^{-1}}^2 \tag{2.2}$$

where $K$ is a diagonal matrix whose elements are $\pi(s)$, $C = \mathbb{E}_{\pi}(\phi_i\phi_i^{\top})$, and $\pi$ is a distribution over the state space $\mathcal{S}$.

Actually, the two objective functions in GTD and GTD2 can be unified as below

$$J(\theta) = \|b - A\theta\|_{M^{-1}}^2, \tag{2.3}$$

---

**Algorithm 1** GTD Algorithms

---

1: **for** $t = 1, \ldots, T$ **do**

2:     Update parameters:    $y_{t+1} = \mathcal{P}_{\mathcal{X}_y}\left(y_t + \alpha_t(\hat{b}_t - \hat{A}_t\theta_t - \hat{M}_t y_t)\right)$    $x_{t+1} = \mathcal{P}_{\mathcal{X}_x}\left(x_t + \alpha_t \hat{A}_t^\top y_t\right)$

3: **end for**

**Output:**    $\tilde{x}_T = \frac{\sum_{t=1}^T \alpha_t x_t}{\sum_{t=1}^T \alpha_t}$      $\tilde{y}_T = \frac{\sum_{t=1}^T \alpha_t y_t}{\sum_{t=1}^T \alpha_t}$

---

where $M = I$ in GTD, $M = C$, in GTD2, $A = \mathbb{E}_\pi[\rho(s, a)\phi(s)(\phi(s) - \gamma\phi(s'))^\top], b = \mathbb{E}_\pi[\rho(s, a)\phi(s)r]$, $\rho(s, a) = \mu(a|s)/\mu_b(a|s)$ is the importance weighting factor. Since the underlying distribution is unknown, we use the data $\mathcal{D} = \{\xi_i = (s_i, a_i, r_i, s'_i)\}_{i=1}^n$ to estimate the value function by minimizing the empirical estimation error, i.e.,

$$\hat{J}(\theta) = 1/T \sum_{i=1}^T \|\hat{b} - \hat{A}\theta\|_{\hat{M}^{-1}}^2$$

where $\hat{A}_i = \rho(s_i, a_i)\phi(s_i)(\phi(s_i) - \gamma\phi(s'_i))^\top, \hat{b}_i = \rho(s_i, a_i)\phi(s_i)r_i, \hat{C}_i = \phi(s_i)\phi(s_i)^\top$.

Liu et al. [2015] derived that the GTD algorithms to minimize (2.3) is equivalent to the stochastic gradient algorithms to solve the following convex-concave saddle point problem

$$\min_x \max_y \left(L(x, y) = \langle b - Ax, y \rangle - \frac{1}{2}\|y\|_M^2\right), \tag{2.4}$$

with $x$ as the parameter $\theta$ in the value function, $y$ as the auxiliary variable used in GTD algorithms. Therefore, we consider the general convex-concave stochastic saddle point problem as below

$$\min_{x \in \mathcal{X}_x} \max_{y \in \mathcal{X}_y} \{\phi(x, y) = \mathbb{E}_\xi[\Phi(x, y, \xi)]\}, \tag{2.5}$$

where $\mathcal{X}_x \subset \mathbb{R}^n$ and $\mathcal{X}_y \subset \mathbb{R}^m$ are bounded closed convex sets, $\xi \in \Xi$ is random variable and its distribution is $\Pi(\xi)$, and the expected function $\phi(x, y)$ is convex in $x$ and concave in $y$. Denote $z = (x, y) \in \mathcal{X}_x \times \mathcal{X}_y \triangleq \mathcal{X}$, the gradient of $\phi(z)$ as $g(z)$, and the gradient of $\Phi(z, \xi)$ as $G(z, \xi)$.

In the stochastic gradient algorithm, the model is updated as: $z_{t+1} = \mathcal{P}_{\mathcal{X}}(z_t - \alpha_t(G(z_t, \xi_t)))$, where $\mathcal{P}_{\mathcal{X}}$ is the projection onto $\mathcal{X}$ and $\alpha_t$ is the step size. After $T$ iterations, we get the model $\tilde{z}_1^T = \frac{\sum_{t=1}^T \alpha_t z_t}{\sum_{t=1}^T \alpha_t}$. The error of the model $\tilde{z}_1^T$ is measured by the primal-dual gap error

$$Err_\phi(\tilde{z}_1^T) = \max_{y \in \mathcal{X}_y} \phi(\tilde{x}_1^T, y) - \min_{x \in \mathcal{X}_x} \phi(x, \tilde{y}_1^T). \tag{2.6}$$

Liu et al. [2015] proved that the estimation error of the GTD algorithms can be upper bounded by their corresponding primal-dual gap error multiply a factor. Therefore, we are going to derive the finite sample primal-dual gap error bound for the convex-concave saddle point problem firstly, and then extend it to the finite sample estimation error bound for the GTD algorithms.

Details of GTD algorithms used to optimize (2.4) are placed in **Algorithm 1**( Liu et al. [2015]).

## 2.2 Related work

The TD algorithms for policy evaluation can be divided into two categories: gradient based methods and least-square(LS) based methods(Dann et al. [2014]). Since LS based algorithms need $\mathcal{O}(d^2)$ storage and computational complexity while GTD algorithms are both of $\mathcal{O}(d)$ complexity, gradient based algorithms are more commonly used when the feature dimension is large. Thus, in this paper, we focus on GTD algorithms.

Sutton et al. [2009a] proposed the gradient-based temporal difference (GTD) algorithm for off-policy policy evaluation problem with linear function approximation. Sutton et al. [2009b] proposed GTD2 algorithm which shows a faster convergence in practice. Liu et al. [2015] connected GTD algorithms to a convex-concave saddle point problem and derive a finite sample bound in both on-policy and off-policy cases for constant step size in i.i.d. setting.

In the realistic Markov setting, although the finite sample bounds for LS-based algorithms have been proved (Lazaric et al. [2012] Tagorti and Scherrer [2015]) LSTD($\lambda$), to the best of our knowledge, there is no previous finite sample analysis work for GTD algorithms.

# 3 Main Theorems

In this section, we will present our main results. In Theorem 1, we present our finite sample bound for the general convex-concave saddle point problem; in Theorem 2, we provide the finite sample bounds for GTD algorithms in both on-policy and off-policy cases. Please refer the complete proofs in the supplementary materials.

Our results are derived based on the following common assumptions(Nemirovski [2004], Duchi et al. [2012], Liu et al. [2015]). Please note that, the bounded-data property in assumption 4 in RL can guarantee the Lipschitz and smooth properties in assumption 5-6 (Please see Propsition 1 ).

**Assumption 1** (Bounded parameter). *There exists $D > 0$, such that $\|z - z'\| \leq D$, $for\ \forall z, z' \in \mathcal{X}$.*

**Assumption 2** (Step size). *The step size $\alpha_t$ is non-increasing.*

**Assumption 3** (Problem solvable). *The matrix $A$ and $C$ in Problem 2.4 are non-singular.*

**Assumption 4** (Bounded data). *Features are bounded by $L$, rewards are bounded by $R_{max}$ and importance weights are bounded by $\rho_{max}$.*

**Assumption 5** (Lipschitz). *For $\Pi$-almost every $\xi$, the function $\Phi(x, y, \xi)$ is Lipschitz for both x and y, with finite constant $L_{1x}, L_{1y}$, respectively. We Denote $L_1 \triangleq \sqrt{2}\sqrt{L_{1x}^2 + L_{1y}^2}$.*

**Assumption 6** (Smooth). *For $\Pi$-almost every $\xi$, the partial gradient function of $\Phi(x, y, \xi)$ is Lipschitz for both x and y with finite constant $L_{2x}, L_{2y}$ respectively. We denote $L_2 \triangleq \sqrt{2}\sqrt{L_{2x}^2 + L_{2y}^2}$.*

For Markov process, the mixing time characterizes how fast the process converge to its stationary distribution. Following the notation of Duchi et al. [2012], we denote the conditional probability distribution $P(\xi_t \in A | \mathcal{F}_s)$ as $P_{[s]}^t(A)$ and the corresponding probability density as $p_{[s]}^t$. Similarly, we denote the stationary distribution of the data generating stochastic process as $\Pi$ and its density as $\pi$.

**Definition 1.** *The mixing time $\tau(P_{[t]}, \eta)$ of the sampling distribution $P$ conditioned on the $\sigma$-field of the initial $t$ sample $\mathcal{F}_t = \sigma(\xi_1, \ldots, \xi_t)$ is defined as: $\tau(P_{[t]}, \eta) \triangleq \inf\left\{\Delta : t \in \mathbb{N}, \int |p_{[t]}^{t+\Delta}(\xi) - \pi(\xi)|d(\xi) \leq \eta\right\}$, where $p_{[t]}^{t+\Delta}$ is the conditional probability density at time $t + \Delta$, given $\mathcal{F}_t$.*

**Assumption 7** (Mixing time). *The mixing times of the stochastic process $\{\xi_t\}$ are uniform. i.e., there exists uniform mixing times $\tau(P, \eta) \leq \infty$ such that, with probability 1, we have $\tau(P_{[s]}, \eta) \leq \tau(P, \eta)$ for all $\eta > 0$ and $s \in \mathbb{N}$.*

Please note that, any time-homogeneous Markov chain with finite state-space and any uniformly ergodic Markov chains with general state space satisfy the above assumption(Meyn and Tweedie [2012]). For simplicity and without of confusion, we will denote $\tau(P, \eta)$ as $\tau(\eta)$.

## 3.1 Finite Sample Bound for Convex-concave Saddle Point Problem

**Theorem 1.** *Consider the convex-concave problem in Eqn (2.5). Suppose Assumption 1,2,5,6 hold. Then for the gradient algorithm optimizing the convex-concave saddle point problem in (2.5), for $\forall \delta > 0$ and $\forall \eta > 0$ such that $\tau(\eta) \leq T/2$, with probability at least $1 - \delta$, we have*

$$Err_\phi(\tilde{z}_1^T) \leq \frac{1}{\sum\limits_{t=1}^{T} \alpha_t}\left[ A + B\sum_{t=1}^{T}\alpha_t^2 + C\tau(\eta)\sum_{t=1}^{T}\alpha_t^2 + F\eta\sum_{t=1}^{T}\alpha_t + H\tau(\eta) \right.$$

$$\left. + 8DL_1\sqrt{2\tau(\eta)\log\frac{\tau(\eta)}{\delta}\left(\sum_{t=1}^{T}\alpha_t^2 + \tau(\eta)\alpha_0\right)} \right]$$

$$where: A = D^2 \quad B = \frac{5}{2}L_1^2 \quad C = 6L_1^2 + 2L_1L_2D \quad F = 2L_1D \quad H = 6L_1D\alpha_0$$

*Proof Sketch of Theorem 1.* By the definition of the error function in (2.6) and the property that $\phi(x, y)$ is convex for $x$ and concave for $y$, the expected error can be bounded as below

$$Err_\phi(\tilde{z}_1^T) \leq \max_z \frac{1}{\sum_{t=1}^{T}\alpha_t}\sum_{t=1}^{T}\alpha_t\left[(z_t - z)^\top g(z_t)\right].$$

Denote $\delta_t \triangleq g(z_t) - G(z_t, \xi_t)$, $\delta'_t \triangleq g(z_t) - G(z_t, \xi_{t+\tau})$, $\delta''_t \triangleq G(z_t, \xi_{t+\tau}) - G(z_t, \xi_t)$. Constructing $\{v_t\}_{t \geq 1}$ which is measurable with respect to $\mathcal{F}_{t-1}$, $v_{t+1} = P_{\mathcal{X}}(v_t - \alpha_t(g(z_t) - G(z_t, \xi_t)))$. We have the following key decomposition to the right hand side in the above inequality, the initiation and the explanation for such decomposition is placed in supplementary materials. For $\forall \tau \geq 0$:

$$\max_z \sum_{t=1}^{T} \alpha_t \left[ (z_t - z)^\top g(z_t) \right] = \max_z \left[ \sum_{t=1}^{T-\tau} \alpha_t \left[ \underbrace{(z_t - z)^\top G(z_t, \xi_t)}_{(a)} + \underbrace{(z_t - v_t)^\top \delta'_t}_{(b)} \right. \right. \tag{3.1}$$

$$\left. \left. + \underbrace{(z_t - v_t)^\top \delta''_t}_{(c)} + \underbrace{(v_t - z)^\top \delta_t}_{(d)} \right] + \underbrace{\sum_{t=T-\tau+1}^{T} \alpha_t \left[ (z_t - z)^\top g(z_t) \right]}_{(e)} \right].$$

For term(a), we split $G(z_t, \xi_t)$ into three terms by the definition of $\mathcal{L}_2$-norm and the iteration formula of $z_t$, and then we bound its summation by $\sum_{t=1}^{T-\tau} \left( \|\alpha_t G(z_t, \xi_t)\|^2 + \|z_t - z\|^2 - \|z_{t+1} - z\|^2 \right)$. Actually, in the summation, the last two terms will be eliminated except for their first and the last terms. Swap the $\max$ and $\sum$ operators and use the Lipschitz Assumption 5, the first term can be bounded. Term (c) includes the sum of $G(z_t, \xi_{t+\tau}) - G(z_t, \xi_t)$, which is might be large in Markov setting. We reformulate it into the sum of $G(z_{t-\tau}, \xi_t) - G(z_t, \xi_t)$ and use the smooth Assumption 6 to bound it. Term (d) is similar to term (a) except that $g(z_t) - G(z_t, \xi_t)$ is the gradient that used to update $v_t$. We can bound it similarly with term (a). Term(e) is a constant that does not change much with $T \to \infty$, and we can bound it directly through upper bound of each of its own terms. Finally, we combine all the upper bounds to each term, use the mixing time Assumption 7 to choose $\tau = \tau(\eta)$ and obtain the error bound in Theorem 1.

We decompose Term(b) into a martingale part and an expectation part. By constructing a martingale difference sequence and using the Azuma's inequality together with the Assumption 7, we can bound Term (b) and finally obtain the high probability error bound. □

**Remark:** (1) With $T \to \infty$, the error bound approaches 0 in order $O(\frac{\sum_{t=1}^{T} \alpha_t^2}{\sum_{t=1}^{T} \alpha_t})$. (2) The mixing time $\tau(\eta)$ will influence the convergence rate. If the Markov process has better mixing property with smaller $\tau(\eta)$, the algorithm converge faster. (3) If the data are i.i.d. generated (the mixing time $\tau(\eta) = 0, \forall \eta$) and the step size is set to the constant $\frac{c}{L_1 \sqrt{T}}$, our bound will reduce to $Err_\phi(\tilde{z}_1^T) \leq \frac{1}{\sum_{t=1}^{T} \alpha_t} \left[ A + B \sum_{t=1}^{T} \alpha_t^2 \right] = \mathcal{O}(\frac{L_1}{\sqrt{T}})$, which is identical to previous work with constant step size in i.i.d. setting (Liu et al. [2015], Nemirovski et al. [2009]). (4) The high probability bound is similar to the expectation bound in the following Lemma 1 except for the last term. This is because we consider the deviation of the data around its expectation to derive the high probability bound.

**Lemma 1.** *Consider the convex-concave problem (2.5), under the same as Theorem 1, we have*

$$\mathbb{E}_{\mathcal{D}}[Err_\phi(\tilde{z}_1^T)] \leq \frac{1}{\sum_{t=1}^{T} \alpha_t} \left[ A + B \sum_{t=1}^{T} \alpha_t^2 + C\tau(\eta) \sum_{t=1}^{T} \alpha_t^2 + F\eta \sum_{t=1}^{T} \alpha_t + H\tau(\eta) \right], \forall \eta > 0,$$

*Proof Sketch of Lemma 1.* We start from the key decomposition (3.1), and bound each term with expectation this time. We can easily bound each term as previously except for Term (b). For term (b), since $(z_t - v_t)$ is not related to $\max$ operator and it is measurable with respect to $\mathcal{F}_{t-1}$, we can bound Term (b) through the definition of mixing time and finally obtain the expectation bound. □

### 3.2 Finite Sample Bounds for GTD Algorithms

As a specific convex-concave saddle point problem, the error bounds in Theorem 1&2 can also provide the error bounds for GTD with the following specifications for the Lipschitz constants.

**Proposition 1.** *Suppose Assumption 1-4 hold, then the objective function in GTD algorithms is Lipschitz and smooth with the following coefficients:*

$$L_1 \leq \sqrt{2}(2D(1+\gamma)\rho_{max}L^2 d + \rho_{max}LR_{max} + \lambda_M)$$
$$L_2 \leq \sqrt{2}(2(1+\gamma)\rho_{max}L^2 d + \lambda_M)$$

*where $\lambda_M$ is the largest singular value of $M$.*

**Theorem 2.** *Suppose assumptions 1-4 hold, then we have the following finite sample bounds for the error $\|V - \tilde{v}_1^T\|_\pi$ in GTD algorithms: In on-policy case, the bound in expectation is $\mathcal{O}\left(\frac{L\sqrt{L^4 d^3 \lambda_M \pi_{max}(1+\tau(\eta))\pi_{max} o_1(T)}}{\nu_C}\right)$ and with probability $1 - \delta$ is*

$$\mathcal{O}\left(\frac{\sqrt{L^4 d^2 \lambda_M \pi_{max}}}{\nu_C}\left(\sqrt{(1+\tau(\eta))L^2 d o_1(T)} + \sqrt{\tau(\eta)\log\left(\frac{\tau(\eta)}{\delta}\right)} o_2(T)\right)\right);$$ *In off-policy case, the bound in expectation is $\mathcal{O}\left(\frac{L^2 d\sqrt{2\lambda_C \lambda_M \pi_{max}(1+\tau(\eta)) o_1(T)}}{\nu_{(A^T M^{-1} A)}}\right)$ and with probability $1 - \delta$ is*

$$\mathcal{O}\left(\frac{\sqrt{2\lambda_C \lambda_M \pi_{max}}}{\nu_{(A^T M^{-1} A)}}\left(\sqrt{L^4 d^2(1+\tau(\eta)) o_1(T)} + \sqrt{\tau(\eta)\log\left(\frac{\tau(\eta)}{\delta}\right)} o_2(T)\right)\right),$$ *where $\nu_C, \nu_{(A^T M^{-1} A)}$ is the smallest eigenvalue of the $C$ and $A^T M^{-1} A$ respectively, $\lambda_C$ is the largest singular value of $C$, $o_1(T) = (\frac{\sum_{t=1}^T \alpha_t^2}{\sum_{t=1}^T \alpha_t})$, $o_2(T) = (\frac{\sqrt{\sum_{t=1}^T \alpha_t^2}}{\sum_{t=1}^T \alpha_t})$.*

We would like to make the following discussions for Theorem 2.

**The GTD algorithms do converge in the realistic Markov setting.** As in Theorem 2, the bound in expectation is $\mathcal{O}\left(\sqrt{(1+\tau(\eta)) o_1(T)}\right)$ and with probability $1 - \delta$ is $\mathcal{O}\left(\sqrt{(1+\tau(\eta)) o_1(T) + \sqrt{\tau(\eta)\log(\frac{\tau(\eta)}{\delta})} o_2(T)}\right)$. If the step size $\alpha_t$ makes $o_1(T) \to 0$ and $o_2(T) \to 0$, as $T \to \infty$, the GTD algorithms will converge. Additionally, in high probability bound, if $\sum_{t=1}^T \alpha_t^2 > 1$, then $o_1(T)$ dominates the order, if $\sum_{t=1}^T \alpha_t^2 < 1$, $o_2(T)$ dominates.

**The setup of the step size can be flexible.** Our finite sample bounds for GTD algorithms converge to 0 if the step size satisfies $\sum_{t=1}^T \alpha_t \to \infty$, $\frac{\sum_{t=1}^T \alpha_t^2}{\sum_{t=1}^T \alpha_t} < \infty$, as $T \to \infty$. This condition on step size is much weaker than the constant step size in previous work Liu et al. [2015], and the common-used step size $\alpha_t = \mathcal{O}(\frac{1}{\sqrt{t}}), \alpha_t = \mathcal{O}(\frac{1}{t}), \alpha_t = c = \mathcal{O}(\frac{1}{\sqrt{T}})$ all satisfy the condition. To be specific, for $\alpha_t = \mathcal{O}(\frac{1}{\sqrt{t}})$, the convergence rate is $\mathcal{O}(\frac{\ln(T)}{\sqrt{T}})$; for $\alpha_t = \mathcal{O}(\frac{1}{t})$, the convergence rate is $\mathcal{O}(\frac{1}{\ln(T)})$, for the constant step size, the optimal setup is $\alpha_t = \mathcal{O}(\frac{1}{\sqrt{T}})$ considering the trade off between $o_1(T)$ and $o_2(T)$, and the convergence rate is $\mathcal{O}(\frac{1}{\sqrt{T}})$.

**The mixing time matters.** If the data are generated from a Markov process with smaller mixing time, the error bound will be smaller, and we just need fewer samples to achieve a fixed estimation error. This finding can explain why the experience replay trick (Lin [1993]) works. With experience replay, we store the agent's experiences (or data samples) at each step, and randomly sample one from the pool of stored samples to update the policy function. By Theorem 1.19 - 1.23 of Durrett [2016], it can be proved that, for arbitrary $\eta > 0$, there exists $t_0$, such that $\forall t > t_0 \max_i |\frac{N_t(i)}{t} - \pi(i)| \le \eta$. That is to say, when the size of the stored samples is larger than $t_0$, the mixing time of the new data process with experience replay is 0. Thus, the experience replay trick improves the mixing property of the data process, and hence improves the convergence rate.

**Other factors that influence the finite sample bound:** (1) With the increasing of the feature norm $L$, the finite sample bound increase. This is consistent with the empirical finding by Dann et al. [2014] that the normalization of features is crucial for the estimation quality of GTD algorithms. (2) With the increasing of the feature dimension $d$, the bound increase. Intuitively, we need more samples for a linear approximation in a higher dimension feature space.

## 4 Experiments

In this section, we report our simulation results to validate our theoretical findings. We consider the general convex-concave saddle problem,

$$\min_x \max_y \left(L(x,y) = \langle b - Ax, y\rangle + \frac{1}{2}\|x\|^2 - \frac{1}{2}\|y\|^2\right) \tag{4.1}$$

where $A$ is a $n \times n$ matrix, b is a $n \times 1$ vector, Here we set $n = 10$. We conduct three experiment and set the step size to $\alpha_t = c = 0.001$, $\alpha_t = \mathcal{O}(\frac{1}{\sqrt{t}}) = \frac{0.015}{\sqrt{t}}$ and $\alpha_t = \mathcal{O}(\frac{1}{t}) = \frac{0.03}{t}$ respectively. In each experiment we sample the data $\hat{A}, \hat{b}$ three ways: sample from two Markov chains with different mixing time but share the same stationary distribution or sample from stationary distribution i.i.d. directly. We sample $\hat{A}$ and $\hat{b}$ from Markov chain by using MCMC Metropolis-Hastings algorithms. Specifically, notice that the mixing time of a Markov chain is positive correlation with the second largest eigenvalue of its transition probability matrix (Levin et al. [2009]), we firstly conduct two transition probability matrix with different second largest eigenvalues( both with 1001 state and the second largest eigenvalue are 0.634 and 0.31 respectively), then using Metropolis-Hastings algorithms construct two Markov chain with same stationary distribution.

We run the gradient algorithm for the objective in (4.1) based on the simulation data, without and with experience replay trick. The primal-dual gap error curves are plotted in Figure 1.

We have the following observations. (1) The error curves converge in Markov setting with all the three setups of the step size. (2) The error curves with the data generated from the process which has small mixing time converge faster. The error curve for i.i.d. generated data converge fastest. (3) The error curve for different step size convergence at different rate. (4) With experience replay trick, the error curves in the Markov settings converge faster than previously. All these observations are consistent with our theoretical findings.

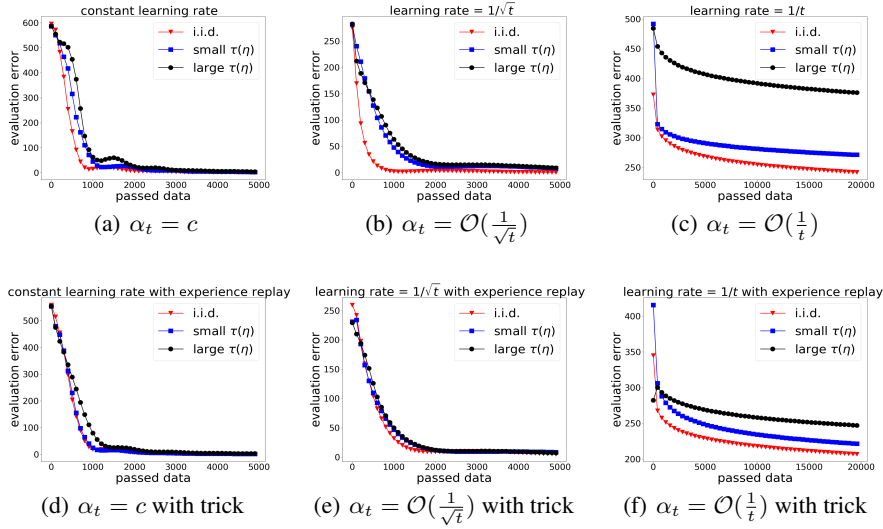

(a) $\alpha_t = c$      (b) $\alpha_t = \mathcal{O}(\frac{1}{\sqrt{t}})$      (c) $\alpha_t = \mathcal{O}(\frac{1}{t})$

(d) $\alpha_t = c$ with trick      (e) $\alpha_t = \mathcal{O}(\frac{1}{\sqrt{t}})$ with trick      (f) $\alpha_t = \mathcal{O}(\frac{1}{t})$ with trick

Figure 1: Experimental Results

## 5 Conclusion

In this paper, in the more realistic Markov setting, we proved the finite sample bound for the convex-concave saddle problems with high probability and in expectation. Then, we obtain the finite sample bound for GTD algorithms both in on-policy and off-policy, considering that the GTD algorithms are specific convex-concave saddle point problems. Our finite sample bounds provide important theoretical guarantee to the GTD algorithms, and also insights to improve them, including how to setup the step size and we need to improve the mixing property of the data like experience replay. In the future, we will study the finite sample bounds for policy evaluation with nonlinear function approximation.

## Acknowledgment

This work was supported by A Foundation for the Author of National Excellent Doctoral Dissertation of RP China (FANEDD 201312) and National Center for Mathematics and Interdisciplinary Sciences of CAS.

## Footnotes

*This work was done when the first author was visiting Microsoft Research Asia.

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
