[Supplementary Material · supp_gtdmc_camera_ready.pdf]

# Finite sample analysis of the GTD Policy Evaluation Algorithms in Markov Settings

This supplementary material gives the detail proof of main theorems in the main paper. The supplementary material is organized as follows:

Section 1 states some assumptions that corresponding to the main paper. Section 2 contains some lemmas and the detail proofs of the main Theorems in the main paper. Section 3 contains detail proofs of lemmas used in proving the theorems.

## 1 Assumptions

**Assumption 1** (Bounded parameter space). *We assume there are finite $D < \infty$ such that*

$$\|z - z'\| \leq D \ \text{ for } z, z' \in \mathcal{X}.$$

**Assumption 2** (Step size). *Let $\{\alpha_t\}$ denote step size sequence which is non-increasing and for $\forall t \quad 0 < \alpha_t \leq \alpha_0 \leq \infty$.*

**Assumption 3** (Problem solvable). *The matrix $A$ and $C$ are non-singular.*

**Assumption 4** (Bounded data). *The max norm of features are bounded by $L$, rewards are bounded by $R_{max}$ and importance weights are bounded by $\rho_{max}$.*

**Assumption 5** (Lipschitz). *For $\Pi$-almost every $\xi$, the function $\Phi(x, y, \xi)$ is Lipschitz for both x and y, that is there exists three constant $0 < L_{1x} < \infty, 0 < L_{1y} < \infty, 0 < L_1 < \infty$ such that:*

$$|\Phi(x', y, \xi) - \Phi(x, y, \xi)| \leq L_{1x}\|x - x'\| \ \text{ for } \ \forall x, x' \in \mathcal{X}_x$$

$$|\Phi(x, y, \xi) - \Phi(x, y', \xi)| \leq L_{1y}\|y - y'\| \ \text{ for } \ \forall y, y' \in \mathcal{X}_y$$

*and let $L_1 \triangleq \sqrt{2}\sqrt{L_{1x}^2 + L_{1y}^2}$, we have*

$$|\Phi(z, \xi) - \Phi(z', \xi)| \leq L_1\|z - z'\| \ \text{ for } \ \forall z, z' \in \mathcal{X}_x \times \mathcal{X}_y.$$

**Assumption 6** (Smooth). *For $\Pi$-almost every $\xi$, the partial gradient function of $\Phi(x, y, \xi)$ is Lipschitz, that is there exists three constant $0 < L_{2x} < \infty, 0 < L_{2y} < \infty, 0 < L_2 < \infty$ such that:*

$$\|G_x(x', y, \xi) - G_x(x, y, \xi)\| \leq L_{2x}\|x - x'\| \ \text{ for } \ \forall x, x' \in \mathcal{X}_x, \forall y \in \mathcal{X}_y$$

$$\|G_y(x, y, \xi) - G_y(x, y', \xi)\| \leq L_{2y}\|y - y'\| \ \text{ for } \ \forall y, y' \in \mathcal{X}_y, \forall x \in \mathcal{X}_x$$

*Then, let $L_2 \triangleq \sqrt{2}\sqrt{L_{2x}^2 + L_{2y}^2}$, we have*

$$\|G(z, \xi) - G(z', \xi)\| \leq L_2\|z - z'\| \ \text{ for } \ \forall z, z' \in \mathcal{X}_x \times \mathcal{X}_y.$$

**Assumption 7.** *The mixing times of the stochastic process $\{\xi_t\}$ are uniform in the sense that there exist uniform mixing times $\tau(P, \eta)$ such that with probability 1 for all $\eta > 0$ and $s \in \mathbb{N}$*

$$\tau(P_{[s]}, \eta) \leq \tau(P, \eta).$$

## 2 Proofs

### 2.1 Proof of Theorem 1

**Theorem 1.** *Suppose Assumption 1, 2,5,6 hold, then for the gradient algorithm optimizing the convex-concave saddle point problem, For $\forall \delta > 0$, and $\forall \eta > 0$ such that $\tau(\eta) \leq T/2$, with probability at least $1 - \delta$, we have*

$$Err_\phi(\tilde{z}_1^T) \leq \frac{1}{\sum\limits_{t=1}^T \alpha_t} \left[ A + B \sum_{t=1}^T \alpha_t^2 + C\tau(\eta) \sum_{t=1}^T \alpha_t^2 + F\eta \sum_{t=1}^T \alpha_t + H\tau(\eta) + 8DL_1 \sqrt{2\tau(\eta) \log \frac{\tau(\eta)}{\delta} \left( \sum_{t=1}^T \alpha_t^2 + \tau(\eta)\alpha_0 \right)} \right].$$

*where*

$$A = D^2 \qquad B = \frac{5}{2}L_1^2 \qquad C = 6L_1^2 + 2L_1L_2D \qquad F = 2L_1D \qquad H = 6L_1D\alpha_0.$$

***Proof of Theorem 1.*** For the convenience of proof, we introduce a sequence of auxiliary variables $\{v_t\}_{t=1,\ldots,T-\tau}$ that follow the iteration formula below respectively:

$$v_1 = z_1 \qquad\qquad v_{t+1} = P_\mathcal{X}\left(v_t - \alpha_t(g(z_t) - G(z_t, \xi_t))\right) \tag{1}$$

Using the definition of error function and $\tilde{z}_1^T$, we can convert expected error function to a more friendly expression, from which we will start our analysis. Define $\Gamma \triangleq \sum_{t=1}^T \alpha_t$.

$$
\begin{aligned}
Err_\phi(\tilde{z}_1^T) &= Err_\phi(\frac{1}{\Gamma} \sum_{t=1}^T \alpha_t z_t) \\
&= \max_y \phi(\frac{1}{\Gamma} \sum_{t=1}^T \alpha_t x_t, y) - \min_x \phi(x, \frac{1}{\Gamma} \sum_{t=1}^T \alpha_t y_t) \\
&\overset{(1)}{=} \max_y \phi(\frac{1}{\Gamma} \sum_{t=1}^T \alpha_t x_t, y) + \max_x -\phi(x, \frac{1}{\Gamma} \sum_{t=1}^T \alpha_t y_t) \\
&\overset{(2)}{\leq} \max_y \frac{1}{\Gamma} \sum_{t=1}^T \alpha_t \phi(x_t, y) + \max_x -\frac{1}{\Gamma} \sum_{t=1}^T \alpha_t \phi(x, y_t) \\
&= \max_x \max_y \frac{1}{\Gamma} \sum_{t=1}^T \alpha_t \left[ \phi(x_t, y) - \phi(x, y_t) \right] \\
&\overset{(3)}{\leq} \max_x \max_y \frac{1}{\Gamma} \sum_{t=1}^T \alpha_t \left[ (x_t - x)^\top g_x(x_t, y_t) + (y - y_t)^\top g_y(x_t, y_t) \right] \\
&= \max_z \frac{1}{\Gamma} \sum_{t=1}^T \alpha_t \left[ (z_t - z)^\top g(z_t) \right] \tag{2}
\end{aligned}
$$

(1) is a consequence of the Lemma 1.
(2) follows by the convexity and concavity of $\phi(x, y)$ with respect to $x$ and $y$ respectively.
(3) follows by the convexity-concavity once again.

Notice that we can not bound the right hand side of (2) directly because of the $max$ operator and non-i.i.d. setting.

we rewrite $z = \underset{z^*}{\operatorname{argmax}} \frac{1}{\Gamma} \sum_{t=1}^T \alpha_t \left[ (z_t - z^*)^\top g(z_t) \right]$, it can be shown that $z$ is measurable with respect to $\mathcal{F}_T$. More specifically, given $\mathcal{F}_t$, $t \leq T$, $z$ is a random variable that correlated to $z_t, \ldots, z_T$.

Previous work that considers the saddle point problem under i.i.d. setting is easy to obtain the bound because they can utilize the i.i.d. property. In their setting, every sample based stochastic gradient

function $G(z_t, \xi_t)$ is unbiased with respected to the $g(z_t)$. Notice $\mathbb{E}[g(z_t) - G(z_t, \xi_t)|\mathcal{F}_{t-1}] = 0$. However, in our Markov setting this term cannot be arbitrary small.

On the other hand, previous work that considers the Markov setting for convex function minimization problem is also easy to obtain the result since they are not bothered by random variable $z$. In their problem $z$ is a constant rather than a random variable which plays a key role when they try to handle the dependent and biased between sampling distribution.

So we cannot apply existing techniques directly or even combine them trivially.

Now, to bound the right-hand side of (2), we consider the following decomposition, for any $\tau \geq 0$. In order to include special $\tau = 0$ case, we require : if $t1 < t2$, $\sum_{t_2}^{t_1}(\cdot) = 0$

$$
\sum_{t=1}^{T} \alpha_t \left[ (z_t - z)^\top g(z_t) \right]
$$

$$
= \sum_{t=1}^{T-\tau} \alpha_t \left[ (z_t - z)^\top g(z_t) \right] + \sum_{t=T-\tau+1}^{T} \alpha_t \left[ (z_t - z)^\top g(z_t) \right]
$$

$$
= \sum_{t=1}^{T-\tau} \alpha_t \left[ (z_t - z)^\top (G(z_t, \xi_t) - G_x(z_t, \xi_t) + g(z_t)) \right]
$$

$$
+ \sum_{t=T-\tau+1}^{T} \alpha_t \left[ (z_t - z)^\top g(z_t) \right],
$$

Denote $g(z_t) - G(z_t, \xi_t)$ as $\delta_t$, denote $g(z_t) - G(z_t, \xi_{t+\tau})$ as $\delta'_t$, denote $G(z_t, \xi_{t+\tau}) - G(z_t, \xi_t)$ as $\delta''_t$, and recall the definition of $v_t$,

$$
\max_z \sum_{t=1}^{T} \alpha_t \left[ (z_t - z)^\top g(z_t) \right] \tag{3}
$$

$$
= \max_z \left[ \sum_{t=1}^{T-\tau} \alpha_t \left[ \underbrace{(z_t - z)^\top G(z_t, \xi_t)}_{(a)} + \underbrace{(z_t - v_t)^\top \delta'_t}_{(b)} + \underbrace{(z_t - v_t)^\top \delta''_t}_{(c)} + \underbrace{(v_t - z)^\top \delta_t}_{(d)} \right] + \underbrace{\sum_{t=T-\tau+1}^{T} \alpha_t \left[ (z_t - z)^\top g(z_t) \right]}_{(e)} \right].
$$

In order to prove Theorem 1, we need the following 7 lemmas, we prove these lemmas in section 3.

**Lemma 1.**

$$
-\min_x (f(x)) = \max_x (-f(x)). \tag{4}
$$

**Lemma 2.** *Let $\{x_t\}$ be a sequence of elements in $\mathbb{R}^n$, let $\{\Delta_t\}$ be a sequence of element in $\mathbb{R}^n$. Given the iteration formula*

$$
x_{t+1} = x_t + \Delta_t.
$$

*Then for $\forall x^* \in \mathbb{R}^n$ we have*

$$
2\langle x^* - x_t, \Delta_t \rangle = \|\Delta_t\|^2 + \|x_t - x^*\|^2 - \|x_{t+1} - x^*\|^2. \tag{5}
$$

*If the parameter space $\mathcal{X}$ is convex and $\mathcal{X} \subset \mathbb{R}^n$ and using the projection step to constrain $x$ in the parameter space during the optimization path, that is*

$$
x_{t+1} = P_{\mathcal{X}}(x_t + \Delta_t)
$$

$$
Then \quad 2\langle x^* - x_t, \Delta_t \rangle \leq \|\Delta_t\|^2 + \|x_t - x^*\|^2 - \|x_{t+1} - x^*\|^2. \tag{6}
$$

**Lemma 3.** *If $z_t$ follows the iteration formula : $z_{t+1} = \mathcal{P}_{\mathcal{X}}(z_t - \alpha_t(G(z_t, \xi_t)))$, then:*

$$
\|\alpha_{t_1} z_{t_1} - \alpha_{t_2+1} z_{t_2+1}\| \leq D(\alpha_{t_1} - \alpha_{t_2+1}) + L_1(t_2 - t_1)\alpha_{t_1}^2.
$$

**Lemma 4.** *If Assumption 2, 1 and 5 hold,*

$$\max_z \sum_{t=1}^{T-\tau} \alpha_t \left[ (z_t - z)^\top G(z_t, \xi_t) \right] \leq \frac{1}{2} D^2 + \frac{1}{2} L_1^2 \sum_{t=1}^{T-\tau} \alpha_t^2.$$

**Lemma 5.** *If Assumption 2, 1 and 5 hold,*

$$\max_z \sum_{t=1}^{T-\tau} \alpha_t (v_t - z)^\top \delta_t \leq \frac{1}{2} D^2 + 2L_1^2 \sum_{t=1}^{T-\tau} \alpha_t^2.$$

**Lemma 6.** *Let Assumption1, 2, 5 and 6 hold and $\delta \in (0,1)$ With probability $1 - \delta$ :*

$$\sum_{t=1}^{T-\tau} \left[ \alpha_t (z_t - v_t)^\top (g(z_t) - G(z_t, \xi_{t+\tau})) \right] \leq 8DL_1 \sqrt{2\tau \log \frac{\tau}{\delta} \left( \sum_{t=1}^{T} \alpha_t^2 + \tau \alpha_0 \right)} + 2DL_1 \sum_{t=1}^{T} \alpha_t \int |\pi(\xi) - p_{[t]}^{t+\tau}(\xi)| d\xi.$$

**Lemma 7.** *If Assumption 2,1, 5 and 6 hold,*

$$\sum_{t=1}^{T-\tau} \left[ \alpha_t (z_t - v_t)^\top (G(z_t, \xi_{t+\tau}) - G(z_t, \xi_t)) \right] \leq 2L_1 \tau (DL_2 + 3L_1) \sum_{t=\tau+1}^{T-\tau} \alpha_{t-\tau}^2 + 5\tau DL_1 \alpha_0.$$

Now we are ready to give the proof of Theorem **??** by constructing a novel decomposition and then use above lemmas.

Firstly, we apply Lemma 4 to bound the term (a),
Then, we apply Lemma 6 to bound the term(b),
Then, we apply Lemma 7 to bound the term(c),
Then, we apply Lemma 5 to bound the term(d),

Finally,for term (e), notice that

$$\mathbb{E} \left[ \max_z \sum_{t=T-\tau+1}^{T} \alpha_t \left[ (z_t - z)^\top g(z_t) \right] \right] \leq \tau DL_1 \alpha_0,$$

Combine above five terms, and for a given $\eta$ , we set $\tau = \tau(\eta)$ by Assumption 7, then the following bound hold with probability $1 - \delta$:

$$Err_\phi(\tilde{z}_1^T)$$

$$\leq \frac{1}{\sum_{t=1}^{T} \alpha_t} \left[ D^2 + \frac{5}{2} L_1^2 \sum_{t=1}^{T} \alpha_t^2 + 2DL_1 \eta \sum_{t=1}^{T} \alpha_t + 2L_1 \tau(\eta)(3L_1 + DL_2) \sum_{t=1}^{T} \alpha_t^2 + 6\tau(\eta)DL_1 \alpha_0 \right.$$

$$\left. + 8DL_1 \sqrt{2\tau(\eta) \log \frac{\tau(\eta)}{\delta} \left( \sum_{t=1}^{T} \alpha_t^2 + \tau(\eta)\alpha_0 \right)} \right]$$

$$\leq \frac{1}{\sum_{t=1}^{T} \alpha_t} \left[ A + B \sum_{t=1}^{T} \alpha_t^2 + C\tau(\eta) \sum_{t=1}^{T} \alpha_t^2 + F\eta \sum_{t=1}^{T} \alpha_t + H\tau(\eta) + 8DL_1 \sqrt{2\tau(\eta) \log \frac{\tau(\eta)}{\delta} \left( \sum_{t=1}^{T} \alpha_t^2 + \tau(\eta)\alpha_0 \right)} \right]$$

where

$$A = D^2 \qquad B = \frac{5}{2} L_1^2 \qquad C = 6L_1^2 + 2L_1 L_2 D \qquad F = 2L_1 D \qquad H = 6L_1 D\alpha_0.$$

$\square$

## 2.2 Proof of Lemma 1 in the main paper

**Lemma 1 in the main paper.** Suppose Assumption 1,2,5,6 hold, then for the gradient algorithm optimizing the convex-concave saddle point problem, $\forall \eta > 0$ such that $\tau(\eta) \leq T/2$, we have

$$\mathbb{E}_{\mathcal{D}}[Err_\phi(\tilde{z}_1^T)] \leq \frac{1}{\sum_{t=1}^T \alpha_t} \left[ A + B \sum_{t=1}^T \alpha_t^2 + C\tau(\eta) \sum_{t=1}^T \alpha_t^2 + F\eta \sum_{t=1}^T \alpha_t + H\tau(\eta) \right], \forall \eta > 0,$$

Firstly, we give a key lemma that will be used in the proof of Lemma 1 in the main paper.

**Lemma 8.** *Let Assumption 1,2, 5 and 7 hold, $\tau \geq 0$, recall the definition of $v_t$ :*

$$\mathbb{E} \left[ \sum_{t=1}^{T-\tau} \alpha_t (z_t - v_t)^\top \delta'_t \right] \leq 2DL_1 \sum_{t=1}^{T-\tau} \alpha_t \mathbb{E} \int |\pi(\xi) - p_{[t]}^{t+\tau}(\xi)| d\xi.$$

***Proof of Lemma 1 in the main paper.*** The result can be obtained by replacing the term (b) in decomposition (3) by an expectation upper bound using Lemma 8 and using Lemma 4, 7, 5 once again to bound the rest term. ☐

## 2.3 Proof of theorem 2

**Theorem 2.** *Suppose assumptions 1-4 hold, then we have the following finite sample bounds for the error $\|V - \tilde{v}_1^T\|_\pi$ in GTD algorithms. In on-policy case, the expectation bound is $\mathcal{O}\left( \frac{L\sqrt{L^4 d^3 \lambda_M \pi_{max}(1+\tau(\eta))\pi_{max} o_1}}{\nu_C} \right)$ and the high probability bound is $\mathcal{O}\left( \frac{\sqrt{L^4 d^2 \lambda_M \pi_{max}}}{\nu_C} \left( \sqrt{(1+\tau(\eta))L^2 d o_1} + \log\left(\frac{\tau(\eta)}{\delta}\right) o_2 \right) \right)$; In off-policy case, the expectation bound is $\mathcal{O}\left( \frac{L^2 d\sqrt{2\lambda_C \lambda_M \pi_{max}(1+\tau(\eta))o_1}}{\nu_{(A^T M^{-1} A)}} \right)$ anf the high probability bound is $\mathcal{O}\left( \frac{\sqrt{2\lambda_C \lambda_M \pi_{max}}}{\nu_{(A^T M^{-1} A)}} \left( \sqrt{L^4 d^2 (1+\tau(\eta)) o_1} + \log\left(\frac{\tau(\eta)}{\delta}\right) o_2 \right) \right)$, where $\nu_C$ is the smallest eigenvalue of the $C$ , $o_1 = \left(\frac{\sum_{t=1}^T \alpha_t^2}{\sum_{t=1}^T \alpha_t}\right), o_2 = \left(\frac{\sqrt{\sum_{t=1}^T \alpha_t^2}}{\sum_{t=1}^T \alpha_t}\right)$.*

**Lemma 9.** *Assumption 4, implies Assumption 5,6, and*

$$L_1^2 \leq 2(2\|A\|D + \|b\| + \lambda_M D)^2,$$
$$L_2^2 \leq 2(2\|A\| + \lambda_M)^2.$$

Here we state three results from Liu et al. [2015].

**Lemma 10** (Lemma 2 of Liu et al. [2015])**.** *For $\forall \xi$, the l2-norm of matrix $\hat{A}_t$ and the l2-norm of vector $\hat{b}_t$ are bounded by*

$$\|\hat{A}_t\|_2 \leq (1+\gamma)\rho_{max} L^2 d, \qquad\qquad \|\hat{b}_t\|_2 \leq \rho_{max} L R_{max}. \qquad (7)$$

**Lemma 11** (Proposition 4 of Liu et al. [2015])**.** *Let V be the value of the target policy and $\tilde{v}_1^T = \Phi \tilde{x}_1^T$, where $\tilde{x}_1^T$ is the value function returned by on policy GTD algorithms. Then we have $\|V - \tilde{v}_1^T\|_\pi \leq \frac{1}{1-\gamma}\left( \|V - \Pi V\|_\pi + \frac{L}{\nu_C}\sqrt{2d\lambda_M \pi_{max} Err(\tilde{x}_1^T, \tilde{y}_1^T)} \right)$.*

**Lemma 12** (Proposition 5 of Liu et al. [2015])**.** *Let V be the value of the target policy and $\tilde{v}_1^T = \Phi \tilde{x}_1^T$, where $\tilde{x}_1^T$ is the value function returned by off policy GTD algorithms. Then we have $\|V - \tilde{v}_1^T\|_\pi \leq \frac{1+\gamma\sqrt{\rho_{max}}}{1-\gamma}\|V - \Pi V\|_\pi + \sqrt{\frac{2\lambda_C \lambda_M \pi_{max}}{\nu_{A^\top M^{-1} A}} Err(\tilde{x}_1^T, \tilde{y}_1^T)}$.*

***Proof of Theorem 2.*** Substitute Lemma 10 into Lemma 9 yields Proposition 1 in the main paper. Then using Proposition 1 together with Proposition 4-5 of Liu et al. [2015] we can obtain the Theorem 2.

☐

# 3 Proof of lemmas

***Proof of lemma 1***. Firstly, we will proof $-\min_x(f(x)) \leq \max_x(-f(x))$ :

$$
\begin{aligned}
\forall x \quad &\max(-f(x)) \geq -f(x) \\
&f(x) \geq -\max(-f(x)) \\
\text{so} \quad &\min(f(x)) \geq -\max(-f(x)) \\
&-\min(f(x)) \leq \max(-f(x))
\end{aligned}
\tag{8}
$$

Then, we will proof $-\min_x(f(x)) \geq \max_x(-f(x))$ :

$$
\begin{aligned}
\forall x \quad &\min(f(x)) \geq f(x) \\
&-\min(f(x)) \leq -f(x) \\
\text{so} \quad &-\min_x f(x) \geq \max_x(-f(x))
\end{aligned}
\tag{9}
$$

Combine the above inequality, we get the result (4). □

***Proof of lemma 2***. Using the iteration formula together with the definition of L2-norm, we can get for $\forall x^* \in \mathbb{R}^n$

$$
\begin{aligned}
\|x_{t+1} - x^*\|^2 - \|x_t - x^*\|^2 &= \|x_t + \Delta_t - x^*\|^2 - \|x_t - x^*\|^2 \\
&= \|x_t - x^*\|^2 + 2\langle x_t - x^*, \Delta_t\rangle + \|\Delta_t\|^2 - \|x_t - x^*\|^2
\end{aligned}
$$

So equation 5 can be obtained by moving the second term from right hand to left hand. And inequation 6 hold because projection is a contraction mapping with respect to $\mathcal{L}_2$ norm.

□

***Proof of lemma 3***.

$$
\begin{aligned}
&\|\alpha_{t_1} z_{t_1} - \alpha_{t_2+1} z_{t_2+1}\| \\
=&\left\| \sum_{s=t_1}^{t_2} \alpha_s z_s - \alpha_{s+1} z_{s+1} \right\| \\
\leq&\left\| \sum_{s=t_1}^{t_2} \alpha_s z_s - \alpha_{s+1}(z_s - \alpha_s G(z_s, \xi_s)) \right\| \\
\leq&\left\| \sum_{s=t_1}^{t_2} \alpha_s z_s - \alpha_{s+1} z_s \right\| + \left\| \sum_{s=t_1}^{t_2} (\alpha_{s+1}\alpha_s G(z_s, \xi_s)) \right\| \\
\leq& \sum_{s=t_1}^{t_2} \|\alpha_s - \alpha_{s+1}\| \|z_s\| + \left\| \sum_{s=t_1}^{t_2} (\alpha_{s+1}\alpha_s G(z_s, \xi_s)) \right\| \\
\leq&D \sum_{s=t_1}^{t_2} (\alpha_s - \alpha_{s+1}) + \sum_{s=t_1}^{t_2} (\alpha_{s+1}\alpha_s \|G(z_s, \xi_s)\|) \\
\leq&D(\alpha_{t_1} - \alpha_{t_2+1}) + L_1(t_2 - t_1)\alpha_{t_1}^2.
\end{aligned}
$$

□

**Proof of lemma 4.** Applying Lemma 2 by setting $x_t = z_t$ , $x^* = z$ and $\Delta_t = -\alpha_t G(z_t, \xi_t)$ can get :

$$\sum_{t=1}^{T-\tau} \alpha_t \left[ (z_t - z)^\top G(z_t, \xi_t) \right]$$

$$\leq \sum_{t=1}^{T-\tau} \frac{1}{2} \left[ \|\alpha_t G(z_t, \xi_t)\|^2 + \|z_t - z\|^2 - \|z_{t+1} - z\|^2 \right]$$

$$\leq \frac{1}{2} \|z_1 - z\|^2 + \frac{1}{2} L_1^2 \sum_{t=1}^{T-\tau} \alpha_t^2$$

The statement follows by taking $max$ on the first term directly.

$\square$

**Proof of lemma 5.** The proof is entirely similar to the proof of 4 if we set $x_t = v_t$ , $x^* = z$ and $\Delta_t = -\alpha_t \delta_t$.

$\square$

**Proof of lemma 6.** We construct a family of martingales, each of which we control with high probability. We begin by defining the following random variables

$$A_t \triangleq \alpha_{t-\tau}(z_{t-\tau} - v_{t-\tau})^\top (g(z_{t-\tau}) - G(z_{t-\tau}, \xi_t))$$

$$\sum_{t=\tau+1}^{T} A_t = \sum_{t=1}^{T-\tau} \mathbb{E} \left[ \alpha_t (z_t - v_t)^\top (g(z_t) - G(z_t, \xi_{t+\tau})) \right]$$

Define the filtration of $\sigma-$fields $\mathcal{A}_i^j = \mathcal{F}_{\tau i + j}$ for $j = 1, \ldots, \tau$. Then we can construct $\tau$ sets of martingales $\{B_1^j, B_2^j, \ldots\}$ for $j = 1, 2, \ldots, \tau$ :

$$B_i^j = A_{i\tau+j} - \mathbb{E} \left[ A_{i\tau+j} | \mathcal{A}_{i-1}^j \right]$$

By definition, $B_i^j$ is measurable with respect to $\mathcal{A}_i^j$, and $\mathbb{E}[B_i^j | \mathcal{A}_{i-1}^j] = 0$. So for each j, the sequence $\{B_i^j, i = 1, 2, \ldots\}$ is a martingale difference sequence adapted to the filtration $\mathcal{A}_i^j$. For a fixed $j_0 \in 1, 2, \ldots, \tau$, the index $i$ for martingale sequence $B_i^{j_0}$ can take value from the index set $\mathcal{I}(j_0)$,

$$\mathcal{I}(j) = \begin{cases} \mathcal{I}_1 = \{1, \ldots, \lfloor T/\tau \rfloor + 1\} & if \quad j \leq T - \tau \lfloor T/\tau \rfloor \\ \mathcal{I}_2 = \{1, \ldots, \lfloor T/\tau \rfloor\} & if \quad j > T - \tau \lfloor T/\tau \rfloor \end{cases}$$

$$\sum_{t=\tau+1}^{T} A_t = \sum_{j=1}^{\tau} \sum_{i \in \mathcal{I}(j)} B_i^j + \sum_{t=\tau+1}^{T} \mathbb{E}[A_t | \mathcal{F}_{t-\tau}] \tag{10}$$

Notice

$$|B_i^j| = |A_{i\tau+j} - \mathbb{E} \left[ A_{i\tau+j} | \mathcal{A}_{i-1}^j \right]| \leq |A_{i\tau+j}| + |\mathbb{E} \left[ A_{i\tau+j} | \mathcal{A}_{i-1}^j \right]| \leq 8DL_1 \alpha_{i\tau+j}$$

So applying the triangle inequality and Azuma's inequality, we can bound the martingale difference sequence term of 10 :

$$P \left( \sum_{j=1}^{\tau} \sum_{i \in \mathcal{I}(j)} B_i^j > \gamma \right) \leq \sum_{j=1}^{\tau} P \left( \sum_{i \in \mathcal{I}(j)} B_i^j > \frac{\gamma}{\tau} \right) \leq \sum_{j=1}^{\tau} \exp \left( -\frac{\gamma^2}{128 D^2 L_1^2 \tau^2 \sum_{i \in \mathcal{I}(j)} \alpha_{i\tau}^2} \right)$$

Notice that $\tau(\alpha_{i\tau})^2 \leq \sum_{j=1}^{\tau} (\alpha_{(i-1)\tau+j})^2$. So $\sum_{i \in \mathcal{I}(j)} \tau(\alpha_{i\tau})^2 \leq \sum_{t=1}^{T} \alpha_t^2 + \tau\alpha_0$ for $\forall j$

Setting $\gamma = 8DL_1\sqrt{2\tau \log \frac{\tau}{\delta}(\sum_{t=1}^{T} \alpha_t^2 + \tau\alpha_0)}$, we get

$$P\left(\sum_{j=1}^{\tau} \sum_{i \in \mathcal{I}(j)} B_i^j > 8DL_1\sqrt{2\tau \log \frac{\tau}{\delta}\left(\sum_{t=1}^{T} \alpha_t^2 + \tau\alpha_0\right)}\right) \leq \delta$$

For the last term of 10, recall the Lemma 8

$$|\mathbb{E}[A_t|\mathcal{F}_t - \tau]| \leq 2DL_1 \int |\pi(\xi) - p_{[t]}^{t+\tau}(\xi)|d\xi$$

combine the above bound completes the proof. □

***Proof of lemma 7***.  Rearrange the left hand side of the above inequation, we can get

$$\sum_{t=1}^{T-\tau} \left[\alpha_t(z_t - v_t)^\top (G(z_t, \xi_{t+\tau}) - G(z_t, \xi_t))\right]$$

$$= \underbrace{\sum_{t=1+\tau}^{T-\tau} \left[\alpha_{t-\tau}(z_{t-\tau} - v_{t-\tau})^\top G(z_{t-\tau}, \xi_t) - \alpha_t(z_t - v_t)^\top G(z_t, \xi_t)\right]}_{(f)}$$

$$+ \sum_{t=T-\tau}^{T} \alpha_{t-\tau}(z_{t-\tau} - v_{t-\tau})^\top G(z_{t-\tau}, \xi_t) - \sum_{t=1}^{\tau} \alpha_t(z_t - v_t)^\top G(z_t, \xi_t)$$

Considering the term $(f)$:

$$\sum_{t=1+\tau}^{T-\tau} \left[\alpha_{t-\tau}(z_{t-\tau} - v_{t-\tau})^\top G(z_{t-\tau}, \xi_t) - \alpha_t(z_t - v_t)^\top G(z_t, \xi_t))\right]$$

$$= \sum_{t=1+\tau}^{T-\tau} \alpha_{t-\tau}(z_{t-\tau} - v_{t-\tau})^\top (G(z_{t-\tau}, \xi_t) - G(z_t, \xi_t))$$

$$+ \sum_{t=1+\tau}^{T-\tau} \left[\alpha_{t-\tau}(z_{t-\tau} - v_{t-\tau}) - \alpha_t(z_t - v_t)\right]^\top G(z_t, \xi_t)$$

$$\leq \sum_{t=1+\tau}^{T-\tau} \alpha_{t-\tau}\|z_{t-\tau} - v_{t-\tau}\|\|G(z_{t-\tau}, \xi_t) - G(z_t, \xi_t)\|$$

$$+ \sum_{t=1+\tau}^{T-\tau} (\alpha_{t-\tau}z_{t-\tau} - \alpha_t z_t)^\top G(z_t, \xi_t) + \sum_{t=1+\tau}^{T-\tau} (\alpha_t v_t - \alpha_{t-\tau}v_{t-\tau})^\top G(z_t, \xi_t)$$

$$\overset{(1)}{\leq} L_2 \sum_{t=1+\tau}^{T-\tau} \alpha_{t-\tau}\|z_{t-\tau} - v_{t-\tau}\|\|\sum_{s=t-\tau}^{t-1} (z_s - z_{s+1})\|$$

$$+ L_1 \sum_{t=1+\tau}^{T-\tau} \|\alpha_{t-\tau}z_{t-\tau} - \alpha_t z_t\| + L_1 \sum_{t=1+\tau}^{T-\tau} \alpha_{t-\tau}\|v_t - v_{t-\tau}\|$$

$$\leq 2DL_1L_2\tau \sum_{t=\tau+1}^{T-\tau} \alpha_{t-\tau}^2 + 2L_1^2\tau \sum_{t=1+\tau}^{T-\tau} \alpha_{t-\tau}^2 + 4L_1^2\tau \sum_{t=1+\tau}^{T-\tau} \alpha_{t-\tau}^2 + \tau\alpha_0 DL_1$$

$$\leq 2L_1\tau(3L_1 + DL_2) \sum_{t=\tau+1}^{T-\tau} \alpha_{t-\tau}^2 + \tau\alpha_0 DL_1$$

The first term of (1) can be bounded by using the iteration formula of $z_t$.
The second term of (1) can be bounded by using the Lemma 3.
The third term of (1) can be bounded by using the iteration formula of $v_t$.

In conclusion, we get the Lemma 7

$$\sum_{t=1}^{T-\tau} \left[ \alpha_t(z_t - v_t)(G(z_t, \xi_{t+\tau}) - G(z_t, \xi_t)) \right]$$

$$\leq 2L_1\tau(3L_1 + DL_2) \sum_{t=\tau+1}^{T-\tau} \alpha_{t-\tau}^2 + \tau\alpha_0 DL_1 + 2DL_1 \sum_{t=1}^{\tau} \alpha_t + 2DL_1 \sum_{t=T-\tau}^{T} \alpha_{t-\tau}$$

$$= 2L_1\tau(3L_1 + DL_2) \sum_{t=\tau+1}^{T-\tau} \alpha_{t-\tau}^2 + 5\tau\alpha_0 DL_1.$$

$\square$

***Proof of lemma 8***.

$$\mathbb{E}\left[ (z_t - v_t)^\top \delta_t' | \mathcal{F}_t \right]$$

$$= \mathbb{E}\left[ \alpha_t(z_t - v_t)^\top (g(z_t) - G(z_t, \xi_{t+\tau})) | \mathcal{F}_t \right]$$

$$= (z_t - v_t)^\top \left[ \mathbb{E}(g(z_t) - G(z_t, \xi_{t+\tau})) | \mathcal{F}_t \right]$$

$$= (z_t - v_t)^\top \int G(z_t, \xi)(\pi(\xi) - p_{[t]}^{t+\tau}(\xi)) d\xi$$

$$\leq 2DL_1 \int |\pi(\xi) - p_{[t]}^{t+\tau}(\xi)| d\xi$$

Taking expectation to the above inequation then summarization $t$ from 1 to $T - \tau$ we can get

$$\mathbb{E}\left[ (z_t - v_t)^\top \delta_t' | \mathcal{F}_t \right] \leq 2DL_1 \int |\pi(\xi) - p_{[t]}^{t+\tau}(\xi)| d\xi$$

$$\mathbb{E}\left[ \sum_{t=1}^{T-\tau} \alpha_t(z_t - v_t)^\top \delta_t' \right] \leq 2DL_1 \sum_{t=1}^{T-\tau} \alpha_t \mathbb{E} \int |\pi(\xi) - p_{[t]}^{t+\tau}(\xi)| d\xi.$$

$\square$

***Proof of lemma 10***. The proof is similar to the proof of Proposition 3 in Liu et al. [2015]. Notice that in the specific RL problem setting our convex-concave problem can be written as

$$\min_x \max_y \left( L(x, y) = \langle b - Ax, y \rangle - \frac{1}{2} \|y\|_M^2 \right),$$

the stochastic gradient vector $G(x, y, \xi_t)$ can be written as

$$G(x, y, \xi_t) = \begin{bmatrix} -\hat{A}_t^\top y \\ -(\hat{b}_t - \hat{A}_t x - \hat{M}_t y) \end{bmatrix}.$$

Similar to the Lemma 2 in Liu et al. [2015], by using Assumption 4 and the definition of $\hat{b}$, $\hat{A}_t$, $\hat{M}_t$ we can see, for $\forall \xi_t, \|G(x, y, \xi_t)\|^2 = 2 \cdot (\|\hat{A}_t^\top y\|^2 + \|\hat{b}_t - \hat{A}_t x - \hat{M}_t y\|^2) \leq 2 \cdot (\|A\|^2 D^2 + (\|b\| + (\|A\| + \lambda_M)D)^2) \leq 2 \cdot (2\|A\|D + \|b\| + \lambda_M D)^2$.

So Lipschitz constant $L_1$ can be set to the upper bound of the gradient.

The smooth constant $L_2$ can be set similarly. $2 \cdot (\|\frac{\partial G(x,y,\xi_t)}{\partial x}\|^2 + \|\frac{\partial G(x,y,\xi_t)}{\partial y}\|^2) \leq 2 \cdot ((\|A\| + \lambda_M)^2 + \|A\|^2) \leq 2 \cdot (2\|A\| + \lambda_M)^2$.

$\square$

# References

Bo Liu, Ji Liu, Mohammad Ghavamzadeh, Sridhar Mahadevan, and Marek Petrik. Finite-sample analysis of proximal gradient td algorithms. In *UAI*, pages 504–513. Citeseer, 2015.