[Reviews · NeurIPS 2017]

Reviewer 1



This paper extends the finite sample analysis of GTD algorithm in Liu et al.(2015) to the following perspectives: 1) change from i.i.d sampling to Markov sampling condition, which is more realistic 2) change the stepsize setting 3) discussed the impact of experience replay as a way of accelerating the mixing time of the Markov chain Here are some comments: 1) In proposition 1, it requires to know the smallest eigenvalue of $C$ and ${A^T}{M^{ - 1}}A$, however, in realistic settings, neither $C$ and ${A^T}{M^{ - 1}}A$ are accessible, and only there empirical estimation $\hat{C}$ and ${\hat{A}^T}{\hat{M}^{ - 1}}\hat{A}$ are available. This becomes a weak-point of such analysis. 2) It should be noted that stepsize is not a big contribution compared with Liu et al.(2015). There is sample complexity analysis before by Nemirovski of convex-concave saddle-point with diminishing stepsize (such as Robbins-Monro stepsize). Using constant stepsize with iterative averaging does not imply the analysis can not be applied to diminishing stepsize cases but can provide better convergence rate than using the diminishing stepsize. 3) Since the paper talks about sampling complexity in Markov settings, an expectation bound seems not necessary in the main part of the paper. The reviewer suggests moving Sec. 3.1 to the Appendix, and enrich Sec. 3.2 from at least the following two: 1. proof sketch part. 2. result compared with the high-prob bound in Liu et al.(2015). 4 The reviewer is especially interested to see the sample complexity comparison between the result in the paper and this paper's analysis method (both applied to Markov sampling settings): https://arxiv.org/pdf/1703.05376.pdf It should be noted that GTD/GTD2 can be either single-time-scale or two-time-scale, and TDC is strictly two-time-scale.

Reviewer 2



The paper addresses an existing limitation with finite-sample bounds for GTD style algorithms. Most existing results assume that samples come from an i.i.d process instead of the true Markov process. This paper shows that finite-sample (sublinear) convergence can be achieved even when the samples indeed come from a Markov process. The results apply both in expectation and with high probability. In terms of significance, the paper closes an important gap in the analysis of the GTD style algorithms. It lifts some limitations of previous work. The results unfortunately depend on a number of complex assumptions that are difficult to verify in practice (particularly Assumptions 5 and 6). The analysis also does not seem to offer new insights into how the methods can be improved and or used in practice. The practical impact of this analysis is likely to be limited. The results are clearly relevant to the NIPS community. The approach relies mostly on existing techniques but uses them in a novel way to solve a complex problem. Proving the results requires a number of auxiliary random variables and it is difficult to check its correctness given the space limitations of conference paper. The paper is well written in general. Given the technical nature of the results, the authors provide ample intuition behind the approach, challenges, and the significance of the results. The presentation does suffer from many language and editing issues that would have to be addressed before publication. There are too many grammatical issues to list them, often words with a different meaning than intended are used, there are many missing or extra articles, and spaces before commas. Minor issues: - line 144: The reference here is oddly placed, making the statement confusing - line 157: the references are incorrectly formatted - reply trick instead of replay trick - theoretical founding instead of theoretical finding - para 60-68 is particularly rife with problem -

Reviewer 3



It is well known that the standard TD algorithm widely used in reinforcement learning does not correspond to the gradient of any objective function, and consequently is unstable when combined with any type of function approximation. Despite the success of methods like deep RL, which combines vanilla TD with deep learning, theoretically TD with nonlinear function approximation is demonstrably unstable. Much work on fixing this fundamental flaw in RL has been in vain, till the work on gradient TD methods by Sutton et al. Unfortunately, these methods work, but their analysis was flawed, based on a heuristic derivation of the method. A recent breakthrough by Liu et al. (UAI 2015) showed that gradient TD methods are essentially saddle point methods that are pure gradient methods that optimize not the original gradient TD loss function (which they do not), but rather the saddle point loss function that arises when converting the original loss function into the dual space. This neat trick opened the door to a rigorous finite sample analysis of TD methods, and has finally surmounted the obstacles in the last three decades of work on this problem. Liu et al. developed several variants of gradient TD, including proximal methods that have improved theoretical and experimental performance. This paper is an incremental improvement to Liu et al., in that it undertakes a similar analysis of saddle point TD methods under a more realistic assumption that the samples are derived from a Markov chain (unlike Liu et al., which assumed IID samples). This paper provides an important next step in our understanding of how to build a more rigorous foundation for the currently shaky world of gradient TD methods. The analysis is done thoroughly and it is supplemented with some simple experiments. The work does not address the nonlinear function approximation case at all. All the current work on Atari video games and deep RL is based on nonlinear function approximations. Is there any hope that gradient TD methods will be shown someday to be robust (at least locally) with nonlinear function approximation?